# Onabotulinumtoxin-A: Previous Prophylactic Treatment Might Improve Subsequent Anti-CGRP Monoclonal Antibodies Response in Patients with Chronic Migraine

**DOI:** 10.3390/toxins15120677

**Published:** 2023-11-30

**Authors:** Giulia Ceccardi, Francesca Schiano di Cola, Salvatore Caratozzolo, Michele Di Pasquale, Marco Bolchini, Alessandro Padovani, Renata Rao

**Affiliations:** Department of Care Continuity and Frialty, Neurology Unit, University of Brescia, 25128 Brescia, Italy; giulia.ceccardi@outlook.it (G.C.); alessandro.padovani@unibs.it (A.P.); renaraodoc@gmail.com (R.R.)

**Keywords:** migraine, chronic migraine, onabotulinumtoxin A, CGRP, anti-CGRP monoclonal antibodies, prevention

## Abstract

The aim of the present study was to evaluate whether previous preventive treatment with onabotulinumtoxin-A might influence subsequent clinical response following a switch to anti-CGRP monoclonal antibodies (mAbs). The present retrospective study was conducted at the Headache Centre—Neurology Clinic at the Spedali Civili Hospital of Brescia between November 2018 and May 2023. The primary objective was to assess clinical outcome (monthly headache days (MHDs), monthly migraine days (MMDs), mean analgesics consumption, and clinical disability according to Migraine Disability Assessment (MIDAS)) following three months (T3) of preventive treatment with anti-CGRP mAbs comparing patients who did and those who did not previously receive treatment with Onabotulinumtoxin-A. Moreover, we aimed to evaluate whether the clinical response to anti-CGRP mAbs was affected by the number of previous Onabotulinumtoxin-A administrations. At T3, compared to Onabotulinumtoxin-A naïve patients, patients who previously received Onabotulinumtoxin-A documented fewer MMDs (3.3 ± 3.7 versus 5.2 ± 5.0; *p* = 0.017) and a lower MIDAS score (23.2 ± 20.9 versus 37.4 ± 39.6; *p* = 0.013). Patients who received at least 3 onabotulinumtoxin-A administrations documented, at T3, lower MMDs compared to those who received fewer cycles (respectively, 2.1 ± 2.7 vs. 6.5 ± 4.4; *p* = 0.024). In conclusion, according to our data, previous treatment with onabotulinumtoxin-A might improve subsequent response to anti-CGRP mAbs preventive treatment.

## 1. Introduction

Chronic migraine (CM) is a debilitating disease that affects up to 5% of patients with migraine [1] with a very high social and economic impact, leading to three times higher direct costs than episodic migraine (EM) [2].

The pathophysiology of CM involves both peripheral and central sensitization of the trigeminovascular system [1]. Peripheral sensitization of the primary afferent nociceptive neurons is characterized by a spontaneous response to external subthreshold stimuli to which they usually do not respond, resulting in the activation/upregulation of meningeal nociceptors such as the transient receptor potential (TRP) channels TRPV1, ATP-gated P2X3 receptors, dopaminergic D1 and D2 receptors, serotonergic 5HT1b/1d receptors, and the calcitonin gene-related peptide (CGRP) receptor [3]. Patients with CM express a significantly higher percentage of transient receptor potential vanilloid type-1 receptor (TRPV1) in nerve fibers innervating the walls of scalp arteries. Consequently, the receptor induces the excitation of the trigeminovascular pathway resulting in a massive release of glutamate and neuropeptides, e.g., CGRP, which in turn promote sensitization [2]. Central sensitization is a state where nociceptive neurons in the trigeminocervical complex (TCC) and in the thalamic posterior and ventral posteromedial nuclei exhibit increased excitability, synaptic strength, and enlargement of their receptive fields beyond the original site of inflammation or injury [1,2]. These mechanisms can then become independent and present with self-stimulation.

Onabotulinumtoxin-A is an intramuscularly injected acetylcholine release inhibitor and neuromuscular blocking agent that was the first prophylactic treatment specifically indicated for chronic migraine. It is a well-tolerated and effective treatment, available since 2010 [4]. Its mechanisms of action are various, acting towards a plethora of targets [5]. In particular, it inhibits the release of neuropeptides such as CGRP [6] and decreases the insertion of TRPV1 in the C-type neurons [3,7]. This is possible through the cleavage of SNAP-25 in the intracellular membrane of the synaptic cleft preventing the fusion of synaptic vesicles and the release of neurotransmitters (such as CGRP and substance P) and inhibiting the insertion of TRPV1 into the terminal membrane of sensory first-order neurons [5]. Its effects seem to act both by blocking the activation of unmyelinated meningeal nociceptors stimulated outside the blood–brain barrier and also stimulated inside the blood–barrier (such as the cortical spreading depression). Melo-Carillo et al. in 2018 demonstrated that in a female rat model, treated with an extracranial injection of onabotulinumtoxin-A and then stimulated with cortical spreading depression after 7–14 days, the prolonged firing of the meningeal nociceptors of unmyelinated C-fibres was significantly reduced, but not their percentage of activated nociceptors [8].

Since 2018, a new prophylactic option has been available for migraine prevention: anti-CGRP monoclonal antibodies (mAbs). These molecules have a direct anti-CGRP effect, binding either the actual CGRP molecule or its receptor. Ant-CGRP mAbs (eptinezumab, erenumab, fremanezumab, and galcanezumab) have all proven safe and effective in the prophylaxis of both EM and CM, with or without medication overuse (MO) and comorbidities [9]. Although not yet available and/or refundable in our country, given the similar pharmacodynamics and favorable outcome in randomized clinical trials, oral anti-CGRP receptor antagonists rimegepant and atogepant [10,11] will surely also provide a significant option in CM prophylaxis. Given the different mechanisms and sites of action, as well as their differences regarding central and peripheral sensitization, it might seem more than plausible that a subsequent and/or combined treatment with these two therapies might have a strong synergistic effect [12].

The aim of the present study was to evaluate whether previous preventive treatment with onabotulinumtoxin-A might influence—positively in our hypothesis—subsequent clinical response following a switch to anti-CGRP mAbs, and whether the number of previous Onabotulinumtoxin-A administration might also influence clinical response.

## 2. Results

One-hundred and twenty-eight patients were enrolled, of which 51 (39.9%) previously treated with onabotulinumtoxin-A. The latter group was treated quarterly with 195 U. Moreover, the time between the last administration of onabotulinumtoxin and the first dose of anti-CGRP monoclonal antibodies was of three months.

The sample included 108 women (84.4%) with a mean age of 44.9 (SD10.9) years old and a mean disease duration of 27.1 (SD10.8) years. Ninety-eight patients (76.6%) documented medication overuse. Allodynia was reported by 66 (51.6%) patients.

At baseline, the mean MHD was 23.7 (SD5.7), MMDs was 13.9 (SD8.0), mean MIDAS score was 108.9 (SD 76.1), and mean analgesics consumption was 24.8 (SD 18.8).

Patients previously treated with onabotulinumtoxin-A documented fewer MMDs at T0 compared to those who did not undertake onabotulinumtoxin-A (respectively, 12.6 ± 9.1 vs. 14.8 ± 7.2; *p* = 0.023), a higher frequency of allodynia (respectively, 62.7% vs. 44.2%; *p* = 0.026) and lower frequency of associated symptoms (74.5% vs. 85.7%; *p* = 0.030).

All clinical and demographical data are presented in Table 1.

Concerning the primary endpoint, patients previously treated with onabotulinumtoxin-A versus those who did not receive this previous therapeutic option, at T3 documented lower mean MMDs (respectively, 3.3 ± 3.7 versus 5.2 ± 5.0; *p* = 0.017), a lower pain intensity according to NRS (5.9 ± 1.0 vs. 6.6 ± 2.0; *p* = 0.013) and a lower MIDAS score (23.2 ± 20.9 versus 37.4 ± 39.6; *p* = 0.013)—see Table 2. Analyzing data as the mean difference from T0 to T3, we also documented a significant difference in terms of analgesics consumption between patients previously treated with onabotulinumtoxin-A versus those who did not (respectively, −19.02 ± 22.06 vs. −13.4 ± 9.77; *p* = 0.003).

On average, patients in the onabotulinumtoxin-A group received 5 (range 1–11) previous administration cycles before initiating anti-CGRP mAbs. Fourteen patients discontinued before the third cycle (minimum number to detect a significant clinical response) for personal choice or scarce tolerance to the injections. Those who discontinued following the third treatment cycle did so due to no or partial response or, regardless of response, still documented > 8 migraine days per month.

Patients who received at least 3 onabotulinumtoxin-A administrations documented, at T3, lower MMDs compared to those who received fewer cycles (respectively, 2.1 ± 2.7 vs. 6.5 ± 4.4; *p* = 0.024). Other clinical outcomes were not statistically significant, although a general trend towards a better outcome in patients who completed the three treatment cycles (see Table 3).

Regarding clinical outcome at T3, for all patients in treatment with anti-CGRP mAbs, compared to T0 a significant reduction in MHDs (11.4 ± 8.2 vs. 24.0 ± 5.6; *p* < 0.0001), MMDs (4.4 ± 4.6 vs. 13.6 ± 8.2; *p* < 0.0001), analgesics consumption (8.6 ± 8.0 vs. 25.0 ± 18.6 *p* < 0.0001), pain intensity (vs. 6.3 ± 1.7 vs. 7.7 ± 1.1; *p* < 0.0001), and MIDAS score (30.5 ± 32.6 vs. 106.0 ± 73.1 *p* < 0.0001) was found. Figure 1 shows each variable subdivided into two groups according to the number of onabotulinumtoxin-A administrations (for patients who received the treatment).

After three months of follow-up, 51.6% of the entire sample (66/128) had a >50% reduction in MHDs that corresponded to 62.7% in the subgroup of patients previously treated with onabotulinumtoxin-A and 46.8% in the subgroup of patients that did not undertake onabotulinumtoxin-A (*p* = 0.81).

## 3. Discussion

This retrospective study confirmed data from the literature on the efficacy of anti-CGRP mAb in chronic migraine patients [9,13,14,15,16].

The aim of the present study was to evaluate whether previous onabotulinumtoxin-A preventive treatment, in patients currently in prophylaxis with anti-CGRP mAbs, might influence clinical response to the latter. Indeed, our data documented that, compared to patients that only attempted the oral standard of care, patients who also attempted onabotulinumtoxin-A—especially those who completed at least three treatment cycles—had a better early clinical response to anti-CGRP mAbs. In particular, the difference was observed in terms of migraine days, which seemed to be significantly lower already at baseline, most likely due to the effect of the recent onabotulinumtoxin-A treatment.

Onabotulinumtoxin-A and anti-CGRP mAbs have both proven effective in CM prophylaxis, as well as medication overuse, not only in randomized registrative clinical trials but also in real-life studies [17,18,19,20,21,22,23]. In particular, these two treatments seem to act synergistically [12], with onabotulinumtoxin-A having an indirect blockage effect on the CGRP pathway compared to mAbs. Indeed, various studies documented a significant efficacy of the combination of the two treatments, i.e., anti-CGRP mAbs as an add-on therapy to onabotulinumtoxin-A [20,24,25,26,27,28,29,30,31].

To the best of our knowledge, no previous study assessed how previous onabotulinumtoxin-A affected subsequent response to anti-CGRP mAbs. Previous reports documented a positive outcome for anti-CGRP mAbs in patients who previously failed onabotulinumtoxin-A [25].

Given the present results, it could be hypothesized that onabotulinumtoxin-A might work as a first step toward reversion from chronic to episodic migraine. Once an episodic frequency has been obtained—if obtained—switching to anti-CGRP mAbs might further improve clinical outcome, going from a high- to a low-frequency episodic migraine. This hypothesis is supported by the finding that patients who received more than three treatment cycles documented fewer migraine days compared to those who received a shorter treatment.

Clinical outcome following anti-CGRP mAbs introduction was better in patients who were previously treated with onabotulinumtoxin-A compared to those who only attempted the oral standard of care, although the former documented a higher baseline MMDs and more frequently presented allodynia, a recognized marker of central sensitization [5]. In some ways, our data could also support combined treatment or even a bridge therapy with both onabotulinumtoxin-A and anti-CGRP mAbs.

On the matter, there is no consensus regarding the feasibility and paradigm for such combined treatment, nor evidence regarding anti-CGRP mAb might be more appropriate in combination with onabotulinumtoxin-A. Given the onabotulinumtoxin-A mechanism of action, which eventually results in reduced CGRP release, an anti-CGRP ligand might be the more favorable combination. However, further studies, in particular cross-sectional studies (in order to assess the precise order of introduction and switch of paradigms) are needed.

Similarly, there is no overt consensus regarding the timing of onabotulinumtoxin-A and anti-CGRP mAbs introduction. Most countries in Europe have specific paradigms regarding reimbursement, with anti-CGRP mAbs being reimbursed only following three prophylactic treatments. However, these indications are based on pharmacoecomics, not clinical outcomes. Indeed, the 2022 European Headache Federation guidelines [9] advised anti-CGRP mAbs to be considered a “first-line” treatment for migraine prevention. On the contrary, the 2018 consensus [4] advised for Onabotulinumtoxin-A to be introduced following two to three previous prophylactic failures. This was advised based on expert opinion and mainly due to safety, tolerability, and costs.

This study has several limitations, with the main being the retrospective design and the sample size. Moreover, a longer follow-up might be needed in order to confirm whether data from the first months of treatment is consistent with later responses. It might be of interest also to evaluate the exact clinical trend of all onabotulinumtoxin-A patients who later switched to anti-CGRP mAbs to obtain a better understanding of response predictors.

## 4. Conclusions

According to our data, previous treatment with onabotulinumtoxin-A might improve subsequent response to anti-CGRP mAbs therapy. Thus, in patients with chronic migraine, prophylactic treatment with onabotulinumtoxin-A represents a valid option to be considered also in those patients who are already eligible for anti-CGRP mAbs prophylaxis, as it might improve their overall clinical response.

## 5. Materials and Methods

### 5.1. Standard Protocol Approvals and Patient Consents

This study received approval from the Ethics Standards Committee on Human Experimentation (local ethics committee of the ASST Spedali Civili Hospital, Brescia: NP 3949, approved 10 August 2020). Full written informed consent was required for all participants.

### 5.2. Study Design and Participants

The present work is an observational retrospective study conducted at the Headache Centre—Neurology Clinic at the Spedali Civili Hospital of Brescia between November 2018 and May 2023.

This study included all adult patients with a diagnosis of CM according to the International Classification of Headache Disorders III (ICHD-III) [32] in prophylactic treatment with anti-CGRP monoclonal antibodies (erenumab, fremanezumab, galcanezumab) with a 3-month follow-up in May 2023. Inclusion criteria were the following: documented history of migraine for at least 12 months, headache diary compilation in the 3 months prior to study enrolment, ≥15 migraine days per month for at least 3 months, and ≥3 previous prophylactic failures.

Clinical and demographical information (disease duration, migraine-associated symptoms and severity, triptans response, previous prophylactic treatments) were collected for all patients. Data regarding headache frequency (monthly headache and migraine days—respectively, MHDs and MMDs), analgesics consumption, attacks’ pain intensity (using the Numerical Rating Scale, NRS), and migraine disability (MIDAS score) were collected at baseline (T0) and following three (T3) months of treatment.

### 5.3. Outcome Measures

The primary endpoint of the present study was to assess whether previous preventive treatment with onabotulinumtoxin A might affect subsequent anti-CGRP mAbs response (T3), compared to patients who only attempted oral standard-of-care therapies. Clinical response was evaluated in terms of monthly headache days (MHDs), monthly migraine days (MMDs), mean analgesics consumption, and clinical disability according to Migraine Disability Assessment (MIDAS).

The secondary endpoints were (1) to evaluate whether the clinical response to anti-CGRP mAbs was affected by the number of previous Onabotulinumtoxin-A administrations; and (2) to evaluate the clinical outcome of patients in treatment with anti-CGRP mAbs at T3.

### 5.4. Statistical Analysis

Shapiro–Wilk test and Levene test were used to assess the normality of the distribution and the homogeneity of variance. Continuous variables were described as mean and standard deviation or median and interquartile range as appropriate, and categorical variables were expressed as frequencies and percentages.

An independent *t*-test was conducted to test whether there were statistically significant differences in MMDs/MHDs, analgesics consumption, and migraine disability (MIDAS score) from baseline to T3 in patients who did the treatment compared to those who did not undertake onabotlunimtoxin A pre-mAbs treatment. A repeated measure *t*-test-test was conducted to assess clinical outcome from baseline to T3.

Clinical response (>50% reduction in MHDs at T3 compared to T0) between the two treatment groups was compared using the Spearman correlation coefficient and Chi-square.

Statistical significance was set at *p* < 0.05. Data analyses were carried out with SPSS software (version 22.0; Armonk, NY, USA).

## Figures and Tables

**Figure 1 toxins-15-00677-f001:**
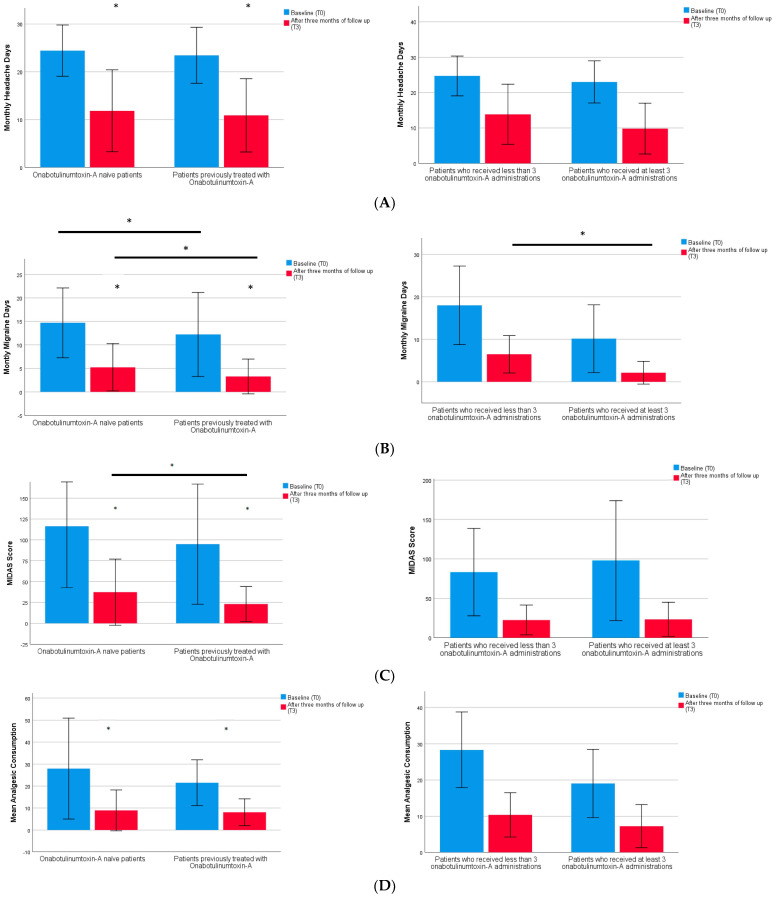
Shows the clinical outcome in patients previously treated with onabotulinumtoxin-A compared to onabotulinumtoxin-A naïve patients at baseline-T0 (blue) and after three months of treatment-T3 (red). Moreover, the graphs on the right show the clinical outcome according to the number of onabotulinumtoxin-A administrations received. In particular, box (**A**) monthly headache days (MHDs), box (**B**) monthly migraine days (MMDs), box (**C**) clinical disability according to Migraine Disability Assessment (MIDAS), box (**D**) mean analgesics consumption, box (**E**) pain intensity. * represent significant *p* value (≤0.05). Bars represent within-subject differences.

**Table 1 toxins-15-00677-t001:** Summary of demographic and clinical characteristics of the entire population before treatment with anti-CGRP mAb. Values shown are numbers with percentages (%) for categorical variables and mean with standard deviation (SD) for continuous variables. * represents significant *p* value (≤0.05).

Demographic and Clinical Characteristics	Entire Sample (n 128)	Patients Previously Treated with Onabotulinumtoxin-A (n 51)	Onabotulinumtoxin-A Naïve Patients (n 77)	*p* Value
Gender (female)	108 (84.4)	43 (84.3)	65 (84.4)	0.988
Age	44.9 (10.9)	46.3 (10.9)	44.0 (10.9)	0.866
Body mass index	22.6 (3.9)	22.4 (3.4)	22.7 (4.2)	0.168
Years of migraine	27.1 (10.8)	27.70 (10.9)	26.8 (10.7)	0.649
Years of chronic migraine	9.2 (6.7)	9.4 (5.8)	9.1 (7.4)	0.102
Number of analgesic overusers	98 (76.6)	38 (74.5)	60 (77.9)	0.559
Number of triptan responders	84 (65.6)	38 (74.5)	46 (59.7)	0.147
Presence of allodynia	66 (51.6)	32 (62.7)	34 (44.2)	0.026 *
Presence of associated symptoms	104 (81.3)	38 (74.5)	66 (85.7)	0.030 *
Anti-CGRP monoclonal antibody				0.085
Erenumab	55 (43.0)	16 (31.4)	39 (50.6)	
Fremanezumab	23 (18.0)	10 (19.6)	13 (16.9)	
Galcanezumab	50 (39.0)	25 (49.0)	25 (32.5)	
Number of previous prophylactic treatments	3 (0.9)	3.6 (0.9)	2.8 (0.9)	0.470
Monthly headache days	24.0 (5.6)	23.3 (6.1)	23.9 (5.5)	0.162
Monthly migraine days	13.6 (8.2)	12.6 (9.1)	14.8 (7.2)	0.023 *
Analgesic consumption	25.0 (18.6)	22.7 (14.1)	26.3 (21.4)	0.089
Pain intensity (using NRS)	7.7 (1.1)	7.5 (1.0)	7.9 (1.1)	0.793
MIDAS score	106.0 (73.1)	102.5 (82.5)	113.3 (71.6)	0.458

**Table 2 toxins-15-00677-t002:** Table 2 shows the clinical outcome in Onabotulinumtoxin-A naïve patients and patients previously treated with Onabotulinumtoxin-A after three months (T3) of anti-CGRP monoclonal antibodies. * represent significant *p* value (≤0.05).

	Patients Previously Treated with Onabotulinumtoxin-A (n 51)	Onabotulinumtoxin-A Naïve Patients (n 77)	*p* Value
Monthly headache days (MHDs) T3	10.9 (7.7)	11.4 (8.6)	0.451
Monthly migraine days (MMDs) T3	3.3 (3.7)	5.2 (5.0)	0.017 *
Pain intensity (using NRS) T3	5.9 (1.0)	6.6 (2.0)	0.013 *
Mean analgesic consumption T3	8.1 (6.1)	8.9 (9.3)	0.105
MIDAS score T3	23.3 (21.0)	37.4 (39.6)	0.013 *

**Table 3 toxins-15-00677-t003:** Table 3 shows the clinical outcome at T3 in patients who received at least three onabotulinumtoxin-A administrations versus patients who received less than three administrations. * represent significant *p* value (≤ 0.05).

	Patients Who Received at Least 3 Onabotulinumtoxin-A Administrations (n 37)	Patients Who Received Less than 3 Onabotulinumtoxin-A Administrations (n 14)	*p* Value
Monthly headache days (MHDs)	9.8 (7.2)	13.6 (8.5)	0.468
Monthly migraine days (MMDs)	2.1 (2.7)	6.5 (4.4)	0.024 *
Pain intensity (using NRS)	5.7 (1.1)	6.3 (0.9)	0.197
Mean analgesic consumption	7.2 (6.0)	10.4 (6.1)	0.995
MIDAS score	23.4 (21.7)	22.6 (19.0)	0.467

## Data Availability

The data presented in this study are available on request from the corresponding author.

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
