# Peer review of "Onabotulinumtoxin-A: Previous Prophylactic Treatment Might Improve Subsequent Anti-CGRP Monoclonal Antibodies Response in Patients with Chronic Migraine"

_toxins, 2023, doi:10.3390/toxins15120677_

Round 1

Reviewer 1 Report

Comments and Suggestions for Authors

Dear Madam/Sir, 

In the manuscript the authors report that patients receiving previous BoNT-A treatments improves the therapeutic outcome after 3 months of treatment with anti-CGRP or anti CGRP-receptor binding monoclonal antibodies fremanezumab, galcanezumab and erenumumab).  

The authors suggest that there is an interaction between the two treatments, but it is very difficult to make any conclusions about the mechanism of interaction (pharmacodynamic, indirect due to plasticity). Is it possibly due to ongoing activity of BoNT-A relevant synaptic sites due to its prolonged enzymatic activity even if analgesic effect is not present anymore? Or BoNT-A might alter the brain plasticity in prolonged manner even when its enzymatic activity ceases? The enzymatic activity of BoNT-A in the periphery (autonomic>muscular>sensory) and CNS (animal models, after axonal transport into CNS in rodents) may last for many months, even up to a year. In regard to this, what is the average time between the last BoNT-A treatment and the beginning of anti-CGRP therapy?  

The main suggestion for the authors regarding the data representation is that is very difficult to follow and compare the exact effect of combined treatments by looking at 2 or 3 tables. Can you please connect T0 and T3 data from both BoNT-A naive and BoNT-A-treated groups into a single figure by extending Figure 1?  In addition, the fig 1 columns lack error bars should contain mean +/- SD or mean +/- SEM representation. If all 4 types of data are represented (two groups, two timepoints), then the data may also be analyzed by 2-way repeated measurement analysis of variance, with post hoc comparisons for individual time points.   

Minor: Figure 1 seems to contain error regarding the NRS pain score (T3 value seems T0 according to tables and vice versa).

Author Response

Dear reviewer,

Thanks for the time and effort you put into the revision of our work.

We have now included the information regarding the time-difference between the last BonT-A administration and anti-CGRP introduction. We have also modified and corrected all figures.

Regarding the stats, I checked with our colleagues form the Statistics Dpt. and he confirmed that the analysis of variance is usually used when more than 2 variables/time points are evaluated, that would not be the case with our data.

Kind regards

Reviewer 2 Report

Comments and Suggestions for Authors

The authors found that a previous treatment with botulinumtoxin A might improve the response to anti CGRP mAB administration in chronic migraine patients. This is an interesting observation:

Comments:

1. There is no information about the dose (standard?) and treatment intervals in the group of patients who had received BoNT before AB.

2. What was the time interval between the last BoNT injection and AB treatment ?

3. At study entry, the group of patients with previous BoNT treatment did significantly better (MMH) than the group with oral medication. Wouldn´t the authors agree that this could have had a marked influence on the results of the study? This point should be discussed.

Comments on the Quality of English Language

.

Author Response

Dear reviewer,

thank you for your comments. We tried to answer your queries as follows:

  1. Q: There is no information about the dose (standard?) and treatment intervals in the group of patients who had received BoNT before AB.1

Answer: We have now added the information in the results section. The standerd dose was 195 IU and the standard treatment interval was three months.

2. Q: What was the time interval between the last BoNT injection and AB treatment ?

Answer: 3 months (we have now added the information in the results section)

3. Q: At study entry, the group of patients with previous BoNT treatment did significantly better (MMH) than the group with oral medication. Wouldn´t the authors agree that this could have had a marked influence on the results of the study? This point should be discussed.

Answer: we believe the previous tretment with BoNTA might indeed have had an effect already at baseline. We discussed it further in the Discussion section. 

Kind regards

Reviewer 3 Report

Comments and Suggestions for Authors

The aim of this paper is to determine if past treatment with onabotulinumoxin-A portends a favorable response to anti-CGRP monoclonal antibodies compared to patients who have not gotten onabotulinumtoxin-A therapy in the past. Overall, the paper is well written and interesting.

In the abstract, the authors state the primary endpoint is: to assess whether previous preventive treatment with onabotulinumtoxin A might 10 affect subsequent anti-CGRP mAbs response compared to patients who only attempted oral standard of care therapies.  This seems more like a primary objective than an endpoint.  The authors should indicate which outcome was used as a primary measure.  Was the primary endpoint monthly headache days or monthly migraine days?

In the introduction, the authors have one paragraph stating that onabotulinumtoxin-A.

is the first therapeutic option with a specific indication for CM- it is not clear why this is listed in one paragraph.  It is better to list out what there is evidence for than “therapeutic prophylactic options” such as topiramate. The author later list CGRP Mabs, but do not mention atogepant which has an indication for chronic migraine. For secondary endpoints, these seem to be secondary aims.

This was done through a retrospective study at an institution in Italy in which multiple modalities were compared over three months, including monthly headache/migraine days in patients started on anti-CGRP monoclonal antibodies. These patients were separated into 2 groups: those previously treated with onabotulinumtoxin-A, and onabotulinumtoxin-A naive patients. The groups differed by the presence of allodynia, associated symptoms and monthly migraine days.

It was found that at 3 months, patients who had been treated with onabotulinumtoxin-A in the past had a lower score in mean monthly migraine days, pain intensity, and MIDAS.

It is not clear when onabotulinumtoxin-A was last used?  Was it 3 months before CGRP Mabs were started?  This should be reported, also in the inclusion.

It would be helpful to show a graph of the two groups, so we can compare trajectories of response including at the 3 injection cycles, compared to usual care.

The study appears novel.  However, it is not a surprise that patients previously treated with onabotulinumtoxin-A injections did better given the indication. 

A more relevant question would be if those benefits persisted. A limitation of this study is the lack of long term follow up to determine persistence (as onabotulinumtoxin-A injections only last 3 months).

The authors provide multiple potential paradigms including combination or bridge; but should discuss the potential benefits of starting CGRP monoclonal antibodies first as a valid clinical option (and the need for cross over studies).

Author Response

Dear reviewer,

We truly appreciate the time and attention spent in revising our work. We tried to assess and answer to all your comments as follows:

The aim of this paper is to determine if past treatment with onabotulinumoxin-A portends a favorable response to anti-CGRP monoclonal antibodies compared to patients who have not gotten onabotulinumtoxin-A therapy in the past. Overall, the paper is well written and interesting.

In the abstract, the authors state the primary endpoint is: to assess whether previous preventive treatment with onabotulinumtoxin A might affect subsequent anti-CGRP mAbs response compared to patients who only attempted oral standard of care therapies.  This seems more like a primary objective than an endpoint.  The authors should indicate which outcome was used as a primary measure.  Was the primary endpoint monthly headache days or monthly migraine days?

We have now rephrased the abstract.

In the introduction, the authors have one paragraph stating that onabotulinumtoxin-A is the first therapeutic option with a specific indication for CM- it is not clear why this is listed in one paragraph.  It is better to list out what there is evidence for than “therapeutic prophylactic options” such as topiramate. The author later list CGRP Mabs, but do not mention atogepant which has an indication for chronic migraine. For secondary endpoints, these seem to be secondary aims.

I have now rephrased the sentence. We wanted to highlight that onabotulinumtoxin-A was the first therapeutic option with a specific indication for chronic migraine (topiramate can be used in episodic migraine as well). That was our focus. I have now included oral anti CGRP receptor antagonists as well (we did not mention them as they are not yet available/reimbursable in our Country). I have also rephrased the final objective statement.

This was done through a retrospective study at an institution in Italy in which multiple modalities were compared over three months, including monthly headache/migraine days in patients started on anti-CGRP monoclonal antibodies. These patients were separated into 2 groups: those previously treated with onabotulinumtoxin-A, and onabotulinumtoxin-A naive patients. The groups differed by the presence of allodynia, associated symptoms and monthly migraine days.

It was found that at 3 months, patients who had been treated with onabotulinumtoxin-A in the past had a lower score in mean monthly migraine days, pain intensity, and MIDAS.

It is not clear when onabotulinumtoxin-A was last used?  Was it 3 months before CGRP Mabs were started?  This should be reported, also in the inclusion.

I have now included this information.

It would be helpful to show a graph of the two groups, so we can compare trajectories of response including at the 3 injection cycles, compared to usual care.

Graphs modified.

The study appears novel.  However, it is not a surprise that patients previously treated with onabotulinumtoxin-A injections did better given the indication.

A more relevant question would be if those benefits persisted. A limitation of this study is the lack of long term follow up to determine persistence (as onabotulinumtoxin-A injections only last 3 months).

We do acknowledge that , in fact we have reported as one of the main limitations.

The authors provide multiple potential paradigms including combination or bridge; but should discuss the potential benefits of starting CGRP monoclonal antibodies first as a valid clinical option (and the need for cross over studies).

I could not find any study/review regarding the benefit of starting anti-CGRP mAbs instead of onabotulinumtoxin-A in patients with CM. I acknowledge that anti-CGRP mAbs have been suggested as a fist line option in migraine prevention by the EHF guidelines (2022) whereas the 2018 consensus on onabutlinumtoxin A suggested its administration following 2-3 previous preventive treatments. I have now extended the discussion on the matter with our opinion on the matter.

Round 2

Reviewer 2 Report

Comments and Suggestions for Authors

.

Comments on the Quality of English Language

.

Author Response

Dear reviewer,

We have now reviewed the English editing, correcting minor typos/mistakes.

Regards

Reviewer 3 Report

Comments and Suggestions for Authors

The changes are appreciated, and this is a much better version of the paper.

It would be helpful to know what CGRP mABs were used in each group and if there were similar across Botox naïve vs previously administered groups- this could go in the chart with P values.

Given those that previously received Botox were better at baseline, it would be helpful to express mean change in MMDs, etc.  Otherwise, it would be expected that those with lower baselines do better.

Comments on the Quality of English Language

Please fix grammatical errors/typo such as below such as reimboursability.  In the US, we use U, not IU, but I defer to the authors.

Author Response

Dear reviewer,

We have modified the article as per your suggestions.

Regards
